# Are Environmental Interventions Targeting Skin Cancer Prevention among Children and Adolescents Effective? A Systematic Review

**DOI:** 10.3390/ijerph17020529

**Published:** 2020-01-14

**Authors:** K. Thoonen, L. van Osch, H. de Vries, S. Jongen, F. Schneider

**Affiliations:** 1Department of Health Promotion, School CAPHRI, Maastricht University, 6229 ER Maastricht, The Netherlands; liesbeth.vanosch@maastrichtuniversity.nl (L.v.O.); hein.devries@maastrichtuniversity.nl (H.d.V.); francine.schneider@maastrichtuniversity.nl (F.S.); 2Education Support department, University Library Maastricht University, 6211 JH Maastricht, The Netherlands; stefan.jongen@maastrichtuniversity.nl

**Keywords:** skin cancer prevention, environmental interventions, sun protection behaviors, children’s health, adolescent health, health promotion

## Abstract

Skin cancer, which is increasing exceedingly worldwide, is substantially preventable by reducing unprotected exposure to ultraviolet radiation (UVR). Several comprehensive interventions targeting sun protection behaviors among children and adolescents in various outdoor settings have been developed; however, there is a lack of insight on stand-alone effectiveness of environmental elements. To compose future skin cancer prevention interventions optimally, identification of effective environmental components is necessary. Hence, an extensive systematic literature search was conducted, using four scientific databases and one academic search engine. Seven relevant studies were evaluated based on stand-alone effects of various types of environmental sun safety interventions on socio-cognitive determinants, sun protection behaviors, UVR exposure, and incidence of sunburns and nevi. Free provision of sunscreen was most often the environmental component of interest, however showing inconsistent results in terms of effectiveness. Evidence regarding shade provision on shade-seeking behavior was most apparent. Even though more research is necessary to consolidate the findings, this review accentuates the promising role of environmental components in skin cancer prevention interventions and provides directions for future multi-component sun safety interventions targeted at children and adolescents in various outdoor settings.

## 1. Introduction

Melanoma and non-melanoma skin cancers (NMSC) are two of the most rapidly increasing cancer types among white populations [1]. Since the early 1980s, melanoma incidence rates have risen twofold in the United States to even threefold in Europe [2,3,4,5]. On a global level, more than 55,000 people died from melanoma in 2012, with the greatest burden in Europe, the United States, Australia, and New Zealand [6,7]. Continuation of the rising incidence rates of melanoma is predicted until 2022 at least in the United States and among European countries [3], also implying increased health care costs and need for a new skin cancer disease management strategy [8,9]. Even though exposure to ultraviolet radiation (UVR) is important for production of vitamin D [10], sun exposure and sunburns during early childhood are the most important risk factors for developing melanoma in later life [11,12,13,14,15] and should therefore be limited [16,17]. Objective data about the overall time children are exposed to UVR are inconsistent and vary per age group, latitude, and country of origin [18,19,20,21]. Even though insight in specific settings where children spend their outside time nowadays is scarce, children seem to often engage in outdoor activities at (pre)school [22,23], around the house and around the beach or swimming pools [24,25], when playing in outdoor playgrounds [26,27] or in public parks [28]. Compared to adolescents, younger children spend time outside after school and during the weekend to a greater extent [16,22].

The Surgeon General and World Health Organization (WHO) have documented five guidelines to enhance sun protection in a call to action addressing the rapidly increasing skin cancer incidence rates. These behaviors consist of wearing protective clothing, wearing sunglasses and a hat, seeking shade, avoiding peak sunlight hours, and applying sunscreen [29]. Over the years, several types of interventions have been developed to encourage various sun protection behaviors among parents and children. Educational interventions for example, can be individually directed and primarily focus on changing intentional decision-making processes [30], by increasing one’s knowledge, improving socio-cognitive determinants (e.g., attitudes, self-efficacy) and learning skills to perform a certain behavior. Positive effects of such interventions have been shown for different sun protection behaviors [31,32] and on knowledge, attitudes, intentions, and behavior in various settings [31,33] for children and adolescents [34,35].

Although educational interventions have demonstrated positive effects, it is important to acknowledge that behavior can also be automatically triggered by environmental characteristics [36]. This is illustrated by dual process models, which state that behavior can both consciously and automatically be influenced by one’s physical environment, such as by physical adaptations, policy or both [37]. One’s environment can be characterized by both different levels of influence (e.g., micro/family setting or macro/community level) as by different types of the environment (e.g., political, economic, social, or physical). Hence, adapting the environment where children and adolescents are highly exposed to UVR can affect sun protection behavior [37].

Overall evidence for effectiveness of multi-component sun safety interventions, integrating both behavioral and environmental strategies, among children and adolescents in outdoor settings, is restricted [38]. Moreover, insight in effects of autonomous elements of these interventions is lacking and additional research is necessary. Since childhood is an important phase where consolidation of health behavior takes place [39] and life-long sun protection habits can be established [40], effectiveness of these interventions needs further examination.

In conclusion, a comprehensive approach in skin cancer prevention strategies among children, targeting both behavioral and environmental factors, is needed. Yet, in order to compose a mix of effective strategies targeting both types of factors and therefore design future interventions optimally, identifying the effects of separate components targeting these factors is necessary. Hence, the aim of this review was to systematically investigate available literature concerning stand-alone effectiveness of environmental interventions targeting sun protection behaviors among children and adolescents in various outdoor settings.

## 2. Methods

### 2.1. Search Strategy

The PRISMA guidelines for systematic reviews were followed to enhance reproducibility of this study [41]. The formulated research question included study characteristics according to the PICOS tool, affirmed by the Cochrane Collaboration [42]. Development of the search strategies was accomplished with help of a scientific information specialist (author S.J.). Since the setting where an intervention took place was specifically important, ‘setting’ characteristics replaced the standardized ‘comparison’ element out of the standardized PICOS in the search strings. An extensive literature search regarding environmental sun safety interventions was conducted using four databases (i.e., PubMed, PsycInfo, Cochrane, Web of Science) and one academic search engine (i.e., Google Scholar) applying systematic formulated search strings. The search strings contained five index terms: (1) population, (2) intervention, (3) setting, (4) outcomes, and (5) study design. These databases together covered a broad range of health, behavioral, and social science subjects as well as the ability to search for scholarly literature. Examples of two search strings are depicted in Appendix A.

### 2.2. Eligibility Criteria

Prior formulated inclusion criteria were for studies looking at: (1) stand-alone effects on socio-cognitive determinants, sun safety behaviors, UVR exposure, and sunburn and nevi incidence; (2) physical environmental, policy, economic, and/or socio-cultural interventions [37]; (3) outdoor and school settings; (4) among children aged between 0 and 18 years; (5) including intervention designs with at least one comparison group, were eligible for inclusion. Stand-alone effectiveness of environmental interventions was interpreted as such that the sun safety intervention, whether it represented a single or multi-component intervention, should consist of one separate intervention arm in which the environmental component was exclusively tested. Inclusion and exclusion criteria are depicted in Table 1. Moreover, studies that were conducted before 1990 and were written in non-English were omitted. No filters were used in the databases and the academic search engine since the search strings were optimally and carefully designed and inclusion and exclusion criteria were clear.

### 2.3. Study Selection

The systematic search took place from the 17 August to the 4 October 2018. An updated search to ensure inclusion of all available recent studies was performed on the 17 September 2019. After eliminating duplicate studies according to the method described by Bramer and colleagues [43], title selection of the studies took place by the first (KT) and last (FS) author in the first round. Consensus between KT and FS about selected studies based on titles needed to be reached before continuing to the next stage. In the second round, eligible abstracts were selected. In the third round, full-text articles were selected. When discrepancies between the two researchers were observed based on the title or abstract, the paper was taken to the second or third round in the study selection process. Both KT and FS determined upon final selection of relevant articles that were used for data abstraction and disagreements were discussed. Furthermore, when no consensus about eligibility could be reached, a third researcher (author L.O.) was consulted.

### 2.4. Data Abstraction

Study characteristics and study outcomes were extracted from the selected studies and quality assessments to estimate the risk of bias were performed. Descriptive information about detailed study characteristics, study outcomes and data regarding quality of studies were reviewed independently before abstracting relevant data.

### 2.5. Study Characteristics

A standardized data abstraction form [44] was critically examined and altered regarding specific characteristics of the studies that were selected, in consensus with the study objectives. Characteristics of the selected studies that were abstracted were predominantly formulated based on the PICOS framework and can be found in Table 2. After entirely reading the first included study [45], study characteristics were further specified according to elaborate data that was present in this study.

### 2.6. Study Outcomes

Study outcomes were regarded important for extraction based on previous systematic reviews [32,38] and were related to socio-cognitive determinants, sun safety behaviors, UVR exposure, and health-related outcomes (i.e., reported sunburns and/or melanocytic nevi (i.e., moles)). The behavioral outcomes were based on recommended sun safety guidelines [29,46]. Since occurrence of sunburn and melanocytic nevi are both objectifications of (over)exposure to UVR, these were included as outcome measures [32]. Finally, information about statistical analyses that were conducted, statistical results, reported stand-alone effects of the intervention and if applicable, reported effect sizes were also abstracted in order to gain an overview of relevant study data.

### 2.7. Study Quality and Risk of Bias

The quality of included studies was assessed by using the Quality Assessment Tool for Quantitative Studies from the Effective Public Health Practice Project (EPHPP) [47], which is a validated tool to use in the assessment of quality of studies [48,49]. With regard to assessing the quality of studies included in the review, methodological characteristics were rated as ‘strong’, ‘moderate’, or ‘weak’.

## 3. Results

In total, 1085 articles were found in the five databases and were screened for eligibility. After the de-duplication process, 753 titles remained. After screening of titles and abstracts, 45 articles were eligible for full-text screening. In the full-text phase, three authors were approached to retrieve more information upon deciding whether these articles were eligible for data abstraction, since follow-up data was not present [50,51,52]. The first authors of these studies were contacted to investigate whether data were available. In one case, a study protocol described the methods for conducting a study in which effects of shaded areas in public parks was investigated from 2013 to 2016 [51]. The first author of this article mentioned that the data would be submitted on a short notice and that follow-up results were presented at two conferences. The follow-up data of this study was therefore included in the analysis as a conference abstract. Another study [52] only described baseline data. After contacting the first author, it was mentioned that follow-up data was retrieved and presented at a conference in 2010, however publication was not expected to be established anytime soon. Therefore, the data from this study were included as a conference abstract too [53]. Finally, the follow-up data of one study [50] was eventually excluded in the analysis after contacting the authors, since stand-alone effects could not be distinguished for children or adolescents specifically. Ultimately, from the 45 articles, eight studies that described the stand-alone effects of environmental interventions were qualified for data-abstraction (see Appendix B for the flowchart). After excluding the last article [50], this resulted in a total of seven studies that were used in the data-extraction process. A full overview of study outcomes and results is provided in Table 3.

### 3.1. Study Results

#### 3.1.1. Types of Environmental Components

In most of the studies, an economic intervention [54] was included in an intervention-arm. More specifically, proactive provision of free sunscreen was the most applied environmental component. In three studies free sunscreen was the exclusive addition in the environmental intervention-arm, while in one study sunscreen provision was one of multiple components. In two studies, bottles of sunscreen were provided to parents at school near the end of the school year [55] or daycare center before spring [56]. In one study, sunscreen was handed out to children themselves, at the end of the school year [57]. Lastly, free sunscreen was provided among other factors (portable shade tents, posters, and policy consultation), in which sunscreen dispensers were placed in city parks, community centers and outdoor recreation sites [45]. In another economic intervention, provision of protective clothing was used [53]. The entire intervention was based on arrangement of clothing, hats, as well as swim shirts for children attending daycare centers.

Subsequently, the second most used intervention type was an adaptation in the physical environment, consisting of shade provision in three studies [54]. The effectiveness of shade sails at secondary school sites [58] and purpose built shade and trees covering public park areas [59] were investigated. Moreover, effects of portable tents, besides sunscreen dispensers, were examined in one study [45].

#### 3.1.2. Effects of Environmental Interventions on Socio-Cognitive Determinants

One study described socio-cognitive determinants among other outcomes [57]. In this case, the environmental intervention consisted of free provision of sunscreen. Children showed a significantly greater reduction in their desire to have a tan in comparison with children from the other intervention and control group. No effects were found on knowledge and awareness.

#### 3.1.3. Effects of Environmental Interventions on Sun Safety Behaviors and UVR Exposure

Six studies had one or more sun safety behaviors as outcome(s); UVR exposure was part of the outcome measures in two studies [55,56]. Two studies solely assessed shade-seeking behavior [58,59] and one study focused on sunscreen use [55]. In the remaining studies more than one sun safety behavior was measured [45,56,57]. Parents estimated their children’s execution of sun safety behaviors in most of the studies [45,55,56,57]. In one study, parents and children were both assessed [57]. The effects of different types of environmental interventions on sun safety behaviors were variable. For instance, economic interventions consisting of free sunscreen or sunscreen dispensers did not account for significant improvements in parental or children’s sun safety behaviors or UVR exposure [55,56,57]. Physical changes in the environment, consisting of shaded areas and purposely planted trees, showed positive effects on shade-seeking behavior of children and adolescents in two studies [58,59]. However, in a study where sunscreen dispensers were combined with shaded areas, behavioral effects were absent [45].

#### 3.1.4. Effects of Environmental Interventions on Melanocytic Nevi and Sunburns

One study assessed nevi incidence as exclusive outcome [53], two studies evaluated incidence of nevi among other outcomes [55,56], and one study investigated sunburn incidence [57]. Findings regarding the effects of economic interventions on nevi and sunburn incidence, such as free provision of sunscreen, were contradicting. One study found that children who received free sunscreen had developed less nevi at the end of the intervention period [55], whereas another study found no effects of free sunscreen provision on nevi incidence [56]. Lastly, in the only study that investigated the effects of free sunscreen provision on sunburn incidence, no evidence for effects was found [57]. In the only study that investigated free provision of UV-protective clothing, hats, and swim shirts [53], findings show that children who did not receive the intervention developed higher incidence of moles on their bodies than children in the intervention group. Specific information about the usage of these garments by child care staff or parents was however absent.

### 3.2. Quality Assessment of Studies

Weak coding was most often due to absence of reported controlling for confounding variables in study designs or statistical analyses. Furthermore, data collection methods were coded as weak in two studies, due to the lack of validated questionnaires [45,57]. Drop-out rates were above 40% in two studies, resulting in a weak coding for this sub-item [45,57]. All studies were randomized or cluster controlled trials, except for one non-randomized trial with pre- and post-tests [59]. Two studies guaranteed blinding in which the outcome assessors were not aware of the intervention status of participants [55,56] and in two studies, blinding of assessors was not possible due to observational methods [58,59]. In most studies, respondents were not aware of the research question. In one study, the blinding procedure was not explained [53]. Notably, none of the included studies reported effect sizes between intervention and control groups. Since the included studies were heterogeneous in terms of intervention type, outcomes, and statistical results and specific statistical information was absent in three studies, a pooled effect size was not calculated. See Table 4 for the quality rating per study.

## 4. Discussion

This systematic literature study is the first to examine stand-alone effects of environmental sun safety interventions among children and adolescents on socio-cognitive determinants, sun safety behaviors, UVR exposure, development of nevi, and sunburn incidence. Seven studies were included, showing that free provision of sunscreen (four times), shade supply (three times) and provision of UV-protective clothing and accessories (one time) were the environmental types of interventions implemented. Five studies showed significant effects of environmental components, assessed after one year on average (yet ranging from four months to three years). Positive effects were especially visible on shade-seeking behavior and incidence of nevi; effects on socio-cognitive determinants, other sun safety behaviors, UVR exposure, and sunburn incidence were not evident. Overall, shade provision seemed to show the most encouraging results. Recent reviews about the effects of shade provision on sun safety behaviors among adults show encouraging results as well [60,61,62].

First, in respect of environmental interventions implemented in the studies included, five used an economic component such as free provision of sunscreen [55,56,57], free sunscreen dispensers [45], or free provision of UV-protective clothing and garments [53]. Physical changes in the environment such as supplying shaded areas, portable shade tents, or planted trees were used in three studies [45,58,59]. Furthermore, sunscreen provision was the most frequently mentioned intervention, even though sunscreen use alone is not sufficient for UVR protection [63,64] and is currently considered an additional recommendation besides other sun safety practices [29,65]. Despite the fact that provision of sunscreen is relatively accessible and low-cost, economic interventions such as provision of clothing, hats, and sunglasses warrant further exploration as well [66,67], since these methods seem to be more effective than sunscreen application [63,68].

Second, when looking at the outcomes of the included studies, results showed that most outcomes were based on internationally recommended sun safety behaviors [29,46], in which sunscreen use was regularly the primary behavioral outcome [45,55,56,57], followed by shade-seeking [58,59] and wearing UV-protective clothing [45,56]. The preference of measuring sunscreen use prior to shade-seeking and clothing behavior in the studies is in accordance with a general popularity of sunscreen application [68]. Even though it is encouraging that environmental sun safety interventions assess a variety of behavioral outcomes, results of the included studies show that actual effects of environmental interventions on behavioral change are scarce and were only found for shade-seeking behavior [58,59]. These results accentuate the importance of enhancing other sun protection behaviors besides sunscreen use [29]. Furthermore, effects of environmental interventions on other outcomes such as incidence of nevi were variable. Two studies showed positive results considering incidence of nevi on the body [53,55]. However, in these studies, no differences in UVR exposure between intervention and control groups were reported while UVR exposure is a great predictor for nevi development [23]. Consistent conclusions about effects of environmental interventions on sunscreen use and incidence of nevi can therefore not be drawn.

Thirdly, with regard to the setting in which interventions were implemented, the school setting (meso level) was most often used. This preference for the school setting is in conformity with the increased attention for school health promotion in general [69]. Although the recognized Healthy Schools Approach is emerging internationally, attention for sun protection behavior is often lacking [70,71,72]. The promising results of comprehensive health promoting school programs emphasize the importance of sun safety in school-settings, with an added value of integrated environmental components [73]. However, other settings can also be of considerable importance with regard to enhancing sun safety. As ecological systems theory imply, the macro level is also of great importance when targeting children’s health behavior [74]. The minority of interventions directed at recreational venues (macro level) is therefore noticeable. Particularly, since the amount of time people spent at recreational settings is increasing among Western populations [75,76]. The amount of UVR exposure at these venues is often high [21] while sun protection is regularly lacking during outdoor activities [77,78,79] and no prevention policies are at place. Moreover, children specifically are at high risk of receiving large amounts of UVR at playgrounds due to unavailability of shaded areas, as revealed by a recent study conducted in Germany [80]. These findings accentuate the importance of intervening in recreational settings. Nevertheless, more knowledge about specific locations where children receive the largest amounts of UVR is necessary before important settings for interventions can be defined, since insight in harmful sun exposure patterns is still lacking [81] and adequate measurement is challenging [82].

Although this review showed that economic types of interventions are most often implemented, it would also be advantageous to analyse the effect of other types of interventions as well. For example, policy intervention types were absent, while these strategies seem to gain positive effects on other health behaviors such as food intake and physical activity [83,84]. Examples of these types of interventions could be scheduling outdoor activities outside of UV peak hours [32], regulation of wearing hats and playing indoors or in the shade [85] or increasing the availability of shaded areas [86].

### 4.1. Strenghts and Limitations of the Included Studies

A few strengths of the included studies are worth mentioning. Almost all studies used methodologically strong designs, mostly (cluster) RCT designs, which strengthens the validation of evidence [87]. Furthermore, all studies used a relatively large sample size and drop-out rates were considered low. Most studies used multiple measurements which generated long-term data. Lastly, in the included studies, children of all age groups were represented. Also, a few limitations should be mentioned. Quality components such as selection bias, handling confounders, and handling withdrawals and drop-out were often rated as weak to moderate [47]. Furthermore, most weak ratings were given for controlling for confounders, which was often caused by an abstinence of information reported in the studies. Moreover, in some studies lack of details disabled the possibility to rate the study quality sufficiently, which may have resulted in inaccurate weak ratings. For example, data on statistical analyses performed were missing and contact with the author did not provide sufficient information [57]. These methodological shortcomings together may have affected the validity of the results. Furthermore, no effect sizes were reported which made comparison of statistical impact between studies difficult. Due to heterogeneity, it was not applicable nor eligible to calculate comparative effect sizes between studies [88]. Follow-up period was mixed in the studies reporting significant effects (ranging from three months to three years), thus making it difficult to assess the impact of various interventions. Lastly, even though it is regarded beneficial to include grey literature in a systematic review [89,90], the results extracted from two conference abstracts [53,59] should be interpreted with caution.

### 4.2. Strengths and Limitations of This Review

First, this review was conducted in accordance with common guidelines and with use of the PRISMA statement for performing and reporting systematic reviews [41,42], which includes relevant topics and concepts and enhances reporting of systematic reviews [91]. Furthermore, five databases were used in the literature search which is a considerable amount of sources [88], which has accounted for avoiding missing relevant titles [92]. Moreover, for assessing risk of bias of the included studies, the validated EPHPP tool was used [47,49], which has proven adequate inter-rater agreement [93]. The current systematic review has also a few limitations. First, the amount of eligible studies that specifically met our criteria turned out limited, which complicates comparison between studies. However, since sampling was done systematically in this review with high sensitivity by formulating specific search strings for five databases, the sample size of available studies seems adequate to draw conclusions [92]. Moreover, there is no golden standard stating a minimum of eligible studies to be included in the synthesis of systematic reviews. Lastly, years of publication of the included studies showed a shortage in recent studies conducted, which demonstrates a possible decrease in interest in environmental components in skin cancer prevention interventions.

### 4.3. Recommendations

Considering the results of this systematic review, several recommendations can be made. The large body of evidence for presence of health promoting sun safety interventions in general is positive, since health promoting adaptations in the environment show promising results on various other health behaviors [94,95,96,97], and the call for sun protection encouragement specifically is reported [29]. Moreover, within skin cancer prevention specifically, a recent study revealed that three decennia of dissemination of the multi-component SunSmart program led to significant improvements in various sun protection behaviors [98]. However, to gain knowledge on effectiveness of stand-alone effects of environmental interventions, it is recommended to investigate the effects of isolated components specifically in future studies. Albeit this review showed availability of a large amount of interventions that were initially eligible for inclusion, the absence of exclusively reported results of the environmental component(s) in most cases, restricted inclusion of those studies and therefore the ability to report extensively on the effectiveness of these components.

Second, sun protection behaviors of children themselves, in various settings, should be considered. Since children and adolescents were the target groups for this review, it is convenient that the responsibility of sun safety for children was expected to be among parents and teachers primarily. However, in a recent study [99], we found that children between approximately 11 and 14 years old, increasingly execute sun safety measures themselves, while parental protection towards their children declines. Hence, examination of children’s own sun protection behaviors should certainly be included in future effect studies.

Third, the included studies in this review showed most studies were conducted in countries outside of Europe-inhabiting Caucasian populations, such as Oceania and Northern America, where high doses of ambient UVR exposure are present [6,100]. Since the incidence rates and melanoma risk have started risen earlier in these countries located at a lower latitude [101], it is understandable that skin cancer prevention strategies are already developed to a greater extent. Moreover, societal norms regarding UVR exposure might differ from those in European countries. Since latitudinal differences or seasonal variation can account for differences in need for sun protection strategies [102], more extensive research in which latitudinal differences are taken into account and, ideally performed in countries where skin cancer prevention is not yet normalized, is needed to translate research into practice. Specifically, since the need for skin cancer prevention due to rapidly growing incidence of melanoma in European countries is crucial [9,103].

Fourth, in all studies, subjective measures such as self-report questionnaires and/or observations were used. To further increase validity, application of objective measures of UVR exposure, such as handheld meters and time-stamped dosimetry [104] and wrist worn dosimetry devices [20], is recommended. Especially, the combination of self-reported and personal dosimeter measurements is promising to consider in future studies [105].

In conclusion, this review demonstrated overall positive results of environmental interventions in five of the seven included studies. Among those, shade provision was the most promising and consistent in increasing shade-seeking behavior. However, more research is necessary to investigate the perpetuation of these findings. As supplying shade provides intervention opportunities in various settings, in both schools and public areas, integrating shade provision in sun safety interventions for children is highly recommended. Moreover, future environmental interventions should focus more specifically on micro and macro levels of influence in children’s social environments, such as the home- and recreational setting.

## Figures and Tables

**Table 1 ijerph-17-00529-t001:** Index terms (PICOS), inclusion, and exclusion criteria.

	Population	Intervention	Comparison	Outcomes	Study Design
**Inclusion criteria**	Infants, toddlers, preschool children, children, and adolescents	Environmental adaptations targeted on sun safety behaviors and skin cancer prevention	Interventions that enable assessment of stand-alone effects of environmental adaptations, using a control group	Effectiveness of environmental adaptations on socio-cognitive determinants, sun safety behaviors, UVR exposure, sunburn incidence, and nevi	Randomized Controlled Trials and (quasi-) experimental designs to objectify effects of interventions
**Exclusion criteria**	A target population of adults or elderly with an age of 18 or above, a population in which children could not be differentiated, children with skin diseases, hospitalized children and childhood cancer survivors	Interventions without environmental components and/or educational interventions only	Interventions without a control group and/or combined interventions without exclusively investigating effects of environmental adaptations	Outcome variables not related to socio-cognitive determinants, sun safety behaviors, UVR exposure, sunburn incidence and nevi	Study designs without a comparison group and study protocols

Note. Even though interventions should target children and adolescents, parents, caregivers, or others can be primary subjects of the included studies. This distinction is made in the study characteristics table (Table 2).

**Table 2 ijerph-17-00529-t002:** Characteristics of studies included in the systematic review.

Authors, Year	Country	Target Group, Recruitment	Sample Size and Setting	Design (Intervention Groups, Duration, Randomization)	Intervention Type and Level	Outcomes	Outcome Measurements
Gallagher et al., 2000	Canada	Target group: Elementary school children aged between 6–7 and 9–10 years Respondents: Children and their parents Recruitment: School principals were first approached for study participation. Parents were then asked for informed consent for enrolling their child in the study	6 elementary schools in Vancouver 458 children at baseline 309 children at follow-up (67.5%)	Design: Two-arm randomized trial Intervention groups: 1. Control group (no intervention; 164 children) 2. Sunscreen intervention (145 children) Duration: Three years Baseline (June 1993) Three posttests (end of summer season in 1994, 1995 and May 1996) Randomization: Children were randomized in either control or intervention group by a statistician	Intervention type: Economic Intervention level: Meso The environmental component consisted of provision of a broad spectrum sunscreen bottle (SPF 30), provided at the parents at the end of each school year. Instructions and information about frequency of application and sunscreen amount were included. The control group did not receive an intervention component	Application of sunscreen, number of counted nevi on the body and sun exposure	Nevi incidence was measured by physical examination from physicians and sun exposure was measured with activity-based questionnaires, combined with minimal erythemal dose (MED) information about sky conditions, latitude, and month of the year
Glanz et al., 2000	United States	Target group: Children aged between 6–8 years Respondents: Parents and recreation staff Recruitment: Recreation program managers were approached for meetings and a recruitment package was provided	14 outdoor recreation (‘Summer Fun’) sites in Hawaii 756 parents at baseline 383 parents at posttest (50.6%) 285 parents at follow-up (37.1%)	Design: Three-arm randomized trial Intervention groups: 1. Control group (110 parents) 2. Education intervention (122 parents) 3. Education and environmental intervention (53 parents) Duration: Three months Baseline Posttest, after 6 weeks Follow-up, after three months Randomization: A blocking strategy was used with balancing size and location for randomization.	Intervention type: Physical Intervention level: Meso The environmental component consisted of on-site sunscreen dispensers, portable shade tents, posters, and policy consultations. The control group did not receive an intervention component.	Sun safety behaviors (using sunscreen, wearing a shirt with sleeves, wearing sunglasses, seeking shade and wearing a hat). An average score of these behavioral outcomes was measured and defined as a ‘sun-protection habit index’.	Sun safety behaviors were measured with self-administration surveys for parents and monitoring forms for recreation staff completed.
Barankin et al., 2001	Canada	Target group Children aged between 9–10 years Respondents Children, their parents and teachers Recruitment: E-mails were sent to all public schools in the Thames Valley District School Board	23 Grade 4 classes from 16 public schools in London, Ontario, Canada 509 children at pretest 366 children at posttest (71.9%) 259 children at follow-up (50.9%)	Design: Three-arm randomized controlled trial Intervention groups: 1. Control group (97 children) 2. Standard group (107 children) 3. Enhanced group (55 children) Duration: Four months Pretest (May 1999) Posttest (June 1999) Follow-up (September 1999) Randomization: Intervention groups were based on a first-come-first-served basis, according to teachers’ response to e-mails. The first 16 schools who responded were randomized in the two intervention groups	Intervention type: Economic Intervention level: Meso The environmental component consisted of provision of sunscreen prior to the summer holiday in 1999, combined with information sheets for parents. Both the standard and enhanced group received educational presentations about skin cancer risk and prevention at the schools. The control group received activity books with some sun safety education.	Children’s attitudes and awareness about consequences of excessive sun exposure and tanning, children’s sun safety behaviors (using sunscreen, avoiding midday activities and wearing UV-protective clothing and sunglasses) and incidence of children’s sunburns.	Children’s attitudes, sun safety behaviors and sunburn incidence was measured with surveys for parents, children and teachers
Bauer et al., 2005	Germany	Target group: Children aged between 2–7 years Respondents: Children and their parents Recruitment: Public nursery schools were selected randomly	78 public nursery schools in Stuttgart and Bochum 1887 children at baseline 1232 children at follow-up (68%)	Design: Randomized Controlled Trial Intervention groups: 1. Control group (398 children) 2. Educational group (369 children) 3. Education + sunscreen group (465 children) Duration: Three years Baseline assessment (summer 1998 in Stuttgart and autumn 1998 in Bochum) Final assessment (summer 2001 in Stuttgart and autumn 2001 in Bochum) Randomization: A random allocation computer program was used	Intervention type: Economic Intervention level: Meso The environmental component consisted of a broad-spectrum sunscreen bottle (SPF 25) and instructions on sunscreen use, which was provided to parents yearly. Both the intervention groups received educational letters (3 times a year) with information on sunscreen use and melanoma prevention. The control group received one educational session prior to the intervention period.	The number of nevi incidence, sun exposure at home and during holidays, sunburns, sunscreen use and wearing protective clothing.	Nevi incidence was measured by physical examination from dermatologists, using a standardized protocol for defining and counting nevi. Children’s sun exposure, history of sunburns and sunscreen use was measured with questionnaires for parents.
Dobbinson et al., 2009	Australia	Target group: Adolescents, aged between 12–18 years Respondents: Adolescents Recruitment: E-mails with study aims and requirements were sent to school principals	51 Secondary schools in Melbourne All schools completed the trial	Design: Cluster Randomized Controlled Trial Intervention groups: 1. Control group (26 schools) 2. Intervention group (25 schools) Duration: Two years Pretest, before installation of shade sails (2004/2005) Posttest, after installation of shade sails (2005/2006) Randomization: A study statistician randomly assigned the schools in groups	Intervention type: Physical Intervention level: Micro The environmental component consisted of different sized built shade sails on school sites. The control group did not receive an intervention component.	The mean number of students seeking shade after establishing the shade sails and the mean number of students using alternative sites (shade avoidance).	Shade use was observed by students with digital video cameras and reviewed by research assistants following a protocol
Harrison et al., 2010	Australia	Target group: Children, aged between 0–35 months Respondents: Children Recruitment: Unknown	25 daycare centers 770 children at baseline measurement (89% response) 544 children at follow-up (70.7% response)	Design: Cluster Randomized Controlled Trial Intervention groups 1. Control group 2. Intervention group Duration: Three years Baseline (November 1999) Follow-up (2000, 2001 and July 2002)	Intervention type: Economic Intervention level Micro The environmental component consisted of provision of sun-protective clothing, hats and swim shirts for children in the daycare centers. The control group did not receive an intervention component	The number of nevi prevalence	Nevi prevalence was measured by full-body skin examinations
Dobbinson et al., 2019	Australia	Target groups and respondents: For observations: All park visitors For self-report surveys: Respondents living nearby the parks For focus groups: Park visitors aged > 13 years Recruitment: Local government councils were invited with letters of support, households received surveys and invitations for focus groups and in the parks, signs were displayed to recruit participants for focus groups	6 public parks in socioeconomically disadvantaged areas in Melbourne	Design: Non-randomized pre-post controlled trial Intervention groups: 1. Control parks (no built shade) 2. Intervention parks (built shade) Duration: Three years Pretest (2013–2014) Posttest (2014–2015) Follow-up (2015–2016) Randomization: Parks were non-randomly selected according to existing refurbishment plans	Intervention type: Physical Intervention level: Macro The environmental component consisted of built shade in the intervention parks.	Shade use	Shade use was measured by observing park users, by self-report surveys and focus groups with respondents living nearby the parks

**Table 3 ijerph-17-00529-t003:** Study outcomes and results of included studies in the systematic review.

Authors, Year	Design	Outcomes Related to Socio-Cognitive Determinants	Outcomes Related to Sun Safe Behavior and UVR Exposure	Outcomes Related to Sunburns/Reported Nevi	Statistical Analyses	Statistical Results	Reported Stand-Alone Effects
Gallagher et al., 2000	Randomized controlled trial	N/A	Parental application of broad spectrum sunscreen (SPF 30) on their child and children’s UVR exposure	Number of nevi on the body (left aside the scalp, genital areas, and the backside)	Linear regression models with number of nevi as outcome and various single predictor variables and interaction terms, using a forward-selection algorithm (*p* < 0.10)	Sunscreen use and UVR exposure: No significant differences in sunscreen use were found Number of nevi: Children in the intervention group developed significantly less nevi (respectively median counts of 24.0 and 28.0, *p* = 0.048). The interaction between randomization to the intervention group and degree of nevi was the strongest statistical predictor of newly developed nevi (Estimates (SE); −0.38 (0.17), *p* = 0.03)	Sunscreen and UVR exposure: Children were equally protected by sunscreen in the two groups, with no significant difference in time spent outdoors Number of nevi: Children from the intervention group developed significantly less nevi at the end of the study period
Glanz et al., 2000	Three-arm randomized trial	N/A	Children’s own sun protection behaviors, defined as a sun protection habit index: -Wearing a shirt with sleeves -Wearing sunglasses -Seeking shade -Wearing a hat -Use of sunscreen	N/A	Mixed model analyses of variance, ANOVA	Sun safe behaviors: Sun protection habit index increased in the education (0.20 and *p* < 0.001) and education + environmental intervention (0.19 and *p* < 0.001) compared to the control group (0.06), whereas solely sunscreen use increased in the education intervention group only (0.16 ± 0.08 and *p* < 0.05). Other behaviors: No significant differences were found	No significant differences in outcomes were found between the education and education + environmental intervention
Barankin et al., 2001	Three-arm randomized controlled trial	Children’s attitudes about tanning and awareness about consequences of excessive sun exposure Teacher’s estimation of children’s awareness of consequences of UVR	Children’s own sunscreen appliance and parental sunscreen application (15-30 min prior to going out in the sun, reapplication), avoidance of midday activities, wearing long sleeved shirts and long pants and sunglasses	Number of sunburns in children	Missing data	Children: The enhanced group showed the greatest reduction (*p* < 0.05) in children’s attitude favoring tanning. No significant differences in other outcomes were found Parents: No significant differences were found Teachers: No statistical results were mentioned	Children in the enhanced intervention group had significantly the greatest decrease in tanning favouring attitudes compared to the other groups
Bauer et al., 2005	Randomized controlled trial	N/A	Parental application of sunscreen and putting on protective clothing and children’s UVR exposure	Newly developed melanocytic nevi and sunburn incidence	Chi-Squared tests, analyses of variance and nonparametric Kruskal–Wallis tests were conducted to test for differences between control and intervention groups. Wilcoxon tests, Chi squared test statistics, and Fisher’s exact test were conducted to study two groups at one time	Sunscreen use: There were group differences in children’s sunscreen use (*p* = 0.03), however not present between the two intervention groups Protective clothing: No significant differences were found Nevi: No significant differences were found Spent holidays: There were group differences in weeks spent on holidays (*p* = 0.02), and in holidays spent further away from the equator (*p* = 0.009)	Children in the education + sunscreen group did not use sunscreen nor wore protective clothing more often than children in the other groups. Also, no differences in development of nevi were found Respondents in the environmental intervention group significantly reported lower median numbers of weeks spent on holidays in sunny climates. However, respondents in this group also reported to go on holidays further away from the equator than respondents in the control group
Dobbinson et al., 2009	Cluster randomized controlled trial	N/A	Usage of shaded areas and usage of alternative sites	N/A	Differences in aggregated shade use (mean value) between pre-test and post-test in both conditions were studies with unpaired *t*-tests. Generalized estimating equations with robust standard errors were fitted to the data to test for interaction between specific school differences and sites. Non-aggregated data were used in linear mixed models to test for intra school correlation coefficients	Shade use: The mean change in use of sites between pretest and post-test was higher in the intervention than the control (mean change of 2.67 and −0.03, *p* = 0.011) group Shade avoidance: The mean change in using different sites in the intervention group was greater for the shaded areas than the alternative sites (difference in mean change between sites 2.70, *p* = 0.007). At the control schools, no significant differences were found	Adolescent active use of purpose built shade increased at the intervention schools
Harrison et al., 2010	Cluster randomized controlled trial	N/A	N/A	Incidence of pigmented moles	Missing data: conference paper	The median count of incident moles was higher in the control than the intervention group (respectively 16; range 0–77 versus 12,5; and 0–74, *p* = 0.02). The median incidence of moles per month was also higher in the control than the intervention group (respectively 0.68 and 0.46, *p* = 0.001)	There was significantly less pigmented mole incidence in the intervention group, compared to the control group
Dobbinson et al., 2019	Non-randomized pre-post controlled trial	N/A	Usage of shaded areas	N/A	Missing data: conference paper	Intervention-received analyses showed increased shade use by visitors (*p* = 0.04)	Significantly more people used shade at follow-up at the intervention parks compared to the control parks

**Table 4 ijerph-17-00529-t004:** Study quality of included studies in the systematic review [47,49].

	Selection Bias	Study Design	Confounders	Blinding	Data Collection Method	Withdrawals and Drop-Outs
Gallagher et al., 2000	Moderate	Strong	Weak	Moderate	Strong	Moderate
Glanz et al., 2000	Moderate	Strong	Strong	Moderate	Weak	Weak
Barankin et al., 2001	Strong	Strong	Weak	Moderate	Weak	Weak
Bauer et al., 2005	Strong	Strong	Strong	Strong	Strong	Moderate
Dobbinson et al., 2009	Weak	Strong	Weak	Moderate	Strong	Strong
Harrison et al., 2010	Moderate	Strong	Weak	Weak	Strong	Strong
Dobbinson et al., 2019	Moderate	Moderate	Strong	Moderate	Strong	N/A

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
