# Peer review of "Are Environmental Interventions Targeting Skin Cancer Prevention among Children and Adolescents Effective? A Systematic Review"

_ijerph, 2020, doi:10.3390/ijerph17020529_

Round 1

Reviewer 1 Report

Overall comment: The analysis of sun protection methods seems reasonable. However, it would be very helpful to put the topic in the larger perspective of such things as vitamin D production and its health benefits vs. reduced risk of skin cancer.

Should a statement be added regarding funding for this work?

Tanning is an adaptation to seasonal variations in solar UV doses

Colloquium paper: human skin pigmentation as an adaptation to UV radiation.

Jablonski NG, Chaplin G.

Proc Natl Acad Sci U S A. 2010 May 11;107 Suppl 2:8962-8. 

Interdisciplinary Perspectives on Sun Safety.

Geller AC, Jablonski NG, Pagoto SL, Hay JL, Hillhouse J, Buller DB, Kenney WL, Robinson JK, Weller RB, Moreno MA, Gilchrest BA, Sinclair C, Arndt J, Taber JM, Morris KL, Dwyer LA, Perna FM, Klein WMP, Suls J.

JAMA Dermatol. 2018 Jan 1;154(1):88-92.

Thus, regular sun exposure can reduce the risk of skin cancer and melanoma:

Occupational sunlight exposure and melanoma in the U.S. Navy.

Garland FC, White MR, Garland CF, Shaw E, Gorham ED.

Arch Environ Health. 1990 Sep-Oct;45(5):261-7.

Interesting papers to consider

Sun exposure and skin cancer, and the puzzle of cutaneous melanoma: A perspective on Fears et al. Mathematical models of age and ultraviolet effects on the incidence of skin cancer among whites in the United States. American Journal of Epidemiology 1977; 105: 420-427.

Armstrong BK, Cust AE.

Cancer Epidemiol. 2017 Jun;48:147-156.

This paper should be discussed as the concern regarding UV exposure during childhood may be misplaced. The reviewer (me) frequently developed extensive sunburns followed by a good tan around the ages of 5 to 10 years and never developed skin cancer or melanoma into his 70s.

This study reveals an increase in NMSC risk associated with adulthood residential UV exposure, with no effect for childhood UV exposure.

Impact of residential UV exposure in childhood versus adulthood on skin cancer risk in Caucasian, postmenopausal women in the Women's Health Initiative.

Ransohoff KJ, Ally MS, Stefanick ML, Keiser E, Spaunhurst K, Kapphahn K, Pagoto S, Messina C, Hedlin H, Manson JE, Tang JY.

Cancer Causes Control. 2016 Jun;27(6):817-23. 

Also this one

Sunburn in childhood and increased sun exposure during annual holidays in sunny areas should be avoided. In contrast, outdoor activities in childhood, including soccer and gardening, should be encouraged because they are associated with a lower risk of melanoma formation.

Outdoor activities in childhood: a protective factor for cutaneous melanoma? Results of a case-control study in 271 matched pairs.

Kaskel P, Sander S, Kron M, Kind P, Peter RU, Krähn G.

Br J Dermatol. 2001 Oct;145(4):602-9.

Risk factors for basal cell carcinoma in a Mediterranean population: role of recreational sun exposure early in life.

Corona R, Dogliotti E, D'Errico M, Sera F, Iavarone I, Baliva G, Chinni LM, Gobello T, Mazzanti C, Puddu P, Pasquini P.

Arch Dermatol. 2001 Sep;137(9):1162-8.

Vitamin D can reduce the risk of adverse skin effects from UV exposure:

Vitamin D as a Therapeutic Option for Sunburn: Clinical and Biologic Implications.

Scott JF, Lu KQ.

DNA Cell Biol. 2017 Nov;36(11):879-882. doi: 10.1089/dna.2017.3978. Epub 2017 Oct 24. No abstract available.

Oral Vitamin D Rapidly Attenuates Inflammation from Sunburn: An Interventional Study.

Scott JF, Das LM, Ahsanuddin S, Qiu Y, Binko AM, Traylor ZP, Debanne SM, Cooper KD, Boxer R, Lu KQ.

J Invest Dermatol. 2017 Oct;137(10):2078-2086.

Different regions of the world have different problems. The regions with greatest risk for skin cancer are where people with skin type adapted for life in low UV regions live in high UV regions such as Australia and New Zealand. Thus, it would be useful to discuss this point.

As for sunscreen, using it permits people to stay in the sun longer, and if it does not have strong UVA blockage, increases the risk for melanoma

Is sunscreen use for melanoma prevention valid for all sun exposure circumstances?

Autier P, Boniol M, Doré JF.

J Clin Oncol. 2011 May 10;29(14):e425-6; author reply e427. doi: 10.1200/JCO.2010.34.4275. Epub 2011 Apr 4. No abstract available.

Epidemiological evidence that UVA radiation is involved in the genesis of cutaneous melanoma.

Autier P, Doré JF, Eggermont AM, Coebergh JW.

Curr Opin Oncol. 2011 Mar;23(2):189-96. doi: 10.1097/CCO.0b013e3283436e5d. Review.

Sunscreen abuse for intentional sun exposure.

Autier P.

Br J Dermatol. 2009 Nov;161 Suppl 3:40-5. doi: 10.1111/j.1365-2133.2009.09448.x. Review.

Sunscreen use and increased duration of intentional sun exposure: still a burning issue.

Autier P, Boniol M, Doré JF.

Int J Cancer. 2007 Jul 1;121(1):1-5. Review.

Cutaneous malignant melanoma: facts about sunbeds and sunscreen.

Autier P.

Expert Rev Anticancer Ther. 2005 Oct;5(5):821-33. Review.

Sun exposure and sun protection in young European children: an EORTC multicentric study.

Severi G, Cattaruzza MS, Baglietto L, Boniol M, Doré JF, Grivegnée AR, Luther H, Autier P; European Organization for Research Treatment of Cancer (EORTC) Melanoma Cooperative Group.

Eur J Cancer. 2002 Apr;38(6):820-6.

The need for protection against UV varies with season. Can that be discussed?

Vitamin D should be discussed as the sun is the most important source and reducing UVB

The risks and benefits of sun exposure 2016.

Hoel DG, Berwick M, de Gruijl FR, Holick MF.

Dermatoendocrinol. 2016 Oct 19;8(1):e1248325.

exposure lowers vitamin D status. Perhaps one to three of these papers could be cited.

Vitamin D in Adolescents: A Systematic Review and Narrative Synthesis of Available Recommendations.

Patseadou M, Haller DM.

J Adolesc Health. 2019 Nov 1. pii: S1054-139X(19)30436-7. 

Vitamin D in adolescence: evidence-based dietary requirements and implications for public health policy.

Smith TJ, Tripkovic L, Lanham-New SA, Hart KH.

Proc Nutr Soc. 2018 Aug;77(3):292-301

A predictive model of serum 25-hydroxyvitamin D in UK white as well as black and Asian minority ethnic population groups for application in food fortification strategy development towards vitamin D deficiency prevention.

O'Neill CM, Kazantzidis A, Kiely M, Cox L, Meadows S, Goldberg G, Prentice A, Kift R, Webb AR, Cashman KD.

J Steroid Biochem Mol Biol. 2017 Oct;173:245-252.

Seasonal Changes in Vitamin D-Effective UVB Availability in Europe and Associations with Population Serum 25-Hydroxyvitamin D.

O'Neill CM, Kazantzidis A, Ryan MJ, Barber N, Sempos CT, Durazo-Arvizu RA, Jorde R, Grimnes G, Eiriksdottir G, Gudnason V, Cotch MF, Kiely M, Webb AR, Cashman KD.

Nutrients. 2016 Aug 30;8(9). pii: E533. doi: 10.3390/nu8090533.

Concurrent beneficial (vitamin D production) and hazardous (cutaneous DNA damage) impact of repeated low-level summer sunlight exposures.

Felton SJ, Cooke MS, Kift R, Berry JL, Webb AR, Lam PM, de Gruijl FR, Vail A, Rhodes LE.

Br J Dermatol. 2016 Dec;175(6):1320-1328.

The contributions of adjusted ambient ultraviolet B radiation at place of residence and other determinants to serum 25-hydroxyvitamin D concentrations.

Kelly D, Theodoratou E, Farrington SM, Fraser R, Campbell H, Dunlop MG, Zgaga L.

Br J Dermatol. 2016 May;174(5):1068-78.

The Big Vitamin D Mistake.

Papadimitriou DT.

J Prev Med Public Health. 2017 Jul;50(4):278-281.

Vitamin D deficiency in Europe: pandemic?

Cashman KD, Dowling KG, Škrabáková Z, Gonzalez-Gross M, Valtueña J, De Henauw S, Moreno L, Damsgaard CT, Michaelsen KF, Mølgaard C, Jorde R, Grimnes G, Moschonis G, Mavrogianni C, Manios Y, Thamm M, Mensink GB, Rabenberg M, Busch MA, Cox L, Meadows S, Goldberg G, Prentice A, Dekker JM, Nijpels G, Pilz S, Swart KM, van Schoor NM, Lips P, Eiriksdottir G, Gudnason V, Cotch MF, Koskinen S, Lamberg-Allardt C, Durazo-Arvizu RA, Sempos CT, Kiely M.

Am J Clin Nutr. 2016 Apr;103(4):1033-44. 

Vitamin D in pediatric age: consensus of the Italian Pediatric Society and the Italian Society of Preventive and Social Pediatrics, jointly with the Italian Federation of Pediatricians.

Saggese G, Vierucci F, Prodam F, Cardinale F, Cetin I, Chiappini E, De' Angelis GL, Massari M, Miraglia Del Giudice E, Miraglia Del Giudice M, Peroni D, Terracciano L, Agostiniani R, Careddu D, Ghiglioni DG, Bona G, Di Mauro G, Corsello G.

Ital J Pediatr. 2018 May 8;44(1):51.

Association Between the 25-Hydroxyvitamin D Status and Physical Performance in Healthy Recreational Athletes.

Zeitler C, Fritz R, Smekal G, Ekmekcioglu C.

Int J Environ Res Public Health. 2018 Dec 3;15(12). pii: E2724. doi: 10.3390/ijerph15122724.

Vitamin D Status and Seasonal Variation among Danish Children and Adults: A Descriptive Study.

Hansen L, Tjønneland A, Køster B, Brot C, Andersen R, Cohen AS, Frederiksen K, Olsen A.

Nutrients. 2018 Nov 20;10(11). pii: E1801. doi: 10.3390/nu10111801.

Vitamin D levels in a pediatric population of a primary care centre: a public health problem?

Fernández Bustillo JM, Fernández Pombo A, Gómez Bahamonde R, Sanmartín López E, Gualillo O.

BMC Res Notes. 2018 Nov 8;11(1):801. 

One of Holick’s reviews would be useful. This one is his most highly cited one.

Holick MF.

N Engl J Med. 2007 Jul 19;357(3):266-81. 

Author Response

Response to reviewer 1

We would like to thank the reviewer for valuable comments and suggestions. We feel that these have enabled us to further optimize the paper. Our response to the indicated comments are below. We genuinely hope that the reviewer finds the revised manuscript suitable for publication in the International Journal of Environmental Research and Public Health.

Comment

‘The analysis of sun protection methods seems reasonable. However, it would be very helpful to put the topic in the larger perspective of such things as vitamin D production and its health benefits vs. reduced risk of skin cancer.’

Response: Thank you for this suggestion. The interest in Vitamin D production within the topic of skin cancer prevention is interesting. For this review, though, we solely focus on extensive sun protection behaviors as the main outcomes. Therefore, studies included in this review do not consider Vitamin D production, nor health benefits of UVR exposure in their findings. Including this outcome in our review at this point would greatly interfere with our fundamental research questions and would lead to a serious alteration of the search strategy and data abstraction.

Comment

‘Should a statement be added regarding funding for this work?’

Response: We have added a statement in which we declare that no funding was received for conducting this study.

Comment

‘This paper (Ransohoff et al., 2016) should be discussed as the concern regarding UV exposure during childhood may be misplaced. The reviewer (me) frequently developed extensive sunburns followed by a good tan around the ages of 5 to 10 years and never developed skin cancer or melanoma into his 70s. This study reveals an increase in NMSC risk associated with adulthood residential UV exposure, with no effect for childhood UV exposure.’

Response: Thank you for this suggestion regarding UVR exposure during childhood and the risk of Non-Melanoma Skin Cancers in later life. We have considered the paper of Ransohoff (et al., 2016), and, even though we acknowledge its value, we still believe in the rationale of the current systematic review, which is to primarily focus on children. This focus is not only justified based on their skin sensitivity, but also due to the fact that the childhood period is crucial for acquiring health behaviour habits that will endure in later life, making children an important target group. Since we have chosen to focus on children aged between 0 and 18 years old as target group – which we have substantiated in the introduction -, we are not able to compare our specific findings with results among adults. Therefore, we believe coverage of the effects of UVR exposure during childhood in comparison with UVR exposure during adulthood, is interesting, but not in line with our aim and therefore not included in the discussion of our review.

Comment

‘Different regions of the world have different problems. The regions with greatest risk for skin cancer are where people with skin type adapted for life in low UV regions live in high UV regions such as Australia and New Zealand. Thus, it would be useful to discuss this point.’

Response: Thank you for this valuable comment. Besides mentioning latitudinal differences and therefore variations in UVR exposure among our target group in the introduction, we have added a section in the discussion in which we reflect on incidence numbers (and prevalence history) in different regions. In this paragraph, we have integrated the topic of latitude and seasonal differences as well. We agree with you that until thus far, attention for skin cancer prevention was predominantly in regions like Oceania and North-America. Moreover, in this part, we reflect on the need for intervention development and dissemination in European countries as well, since this expansion of skin cancer prevention strategies is of utmost importance nowadays.

Comment

‘As for sunscreen, using it permits people to stay in the sun longer, and if it does not have strong UVA blockage, increases the risk for melanoma.’

Response: In the discussion, we have indeed pointed out that sunscreen use alone is not a sufficient sun protection method (Linos et al., 2011 and Ghiasvand et al., 2015) and therefore, other sun protection strategies are of great importance to encourage.

Comment

‘The need for protection against UV varies with season. Can that be discussed?’

Response: Thank you for this suggestion. In the discussion, we have integrated a paragraph in which we reflect on latitudinal differences as well as seasonal variation within the scope of skin cancer prevention strategies.

Comment

‘Vitamin D should be discussed as the sun is the most important source and reducing UVB exposure lowers vitamin D status. Perhaps one to three of these papers could be cited.’

Response: Thank you for this suggestion. As we point out responding to your first comment, the specific topic of Vitamin D production interferes with our initial research questions, and is therefore not discussed in the included studies. However, we acknowledge the importance of this matter and have included a statement in the introduction about the importance of Vitamine D production.

Reviewer 2 Report

This review is of some interest although the topic has rarely been covered, resulting in a limited number of studies that could be included, almost all published more than thirteen years ago. Comments follow for consideration by the authors.

 As only seven studies were judged relevant for inclusion by the authors, the review is too long at 35 printed pages and is repetitive in places. It could be considerably shortened, thereby making more impact. Suggestions for this are outlined below for each section. Introduction. This is not sufficiently focussed. Explaining details of dual process models in the fourth paragraph is not appropriate, for example. References in the first paragraph need to be updated. Mistakes in English are frequent such as in line 29 where “one” should be “two”, in line 47 where “recreating” should be “recreation”, “proportional time” in line 48 does not mean anything, nor does “proportional amount” in line 63. What does “micro” or “macro” level mean in line 86? In line 90, the guidelines are those advised by WHO; the UV Index threshold of 3 has been omitted. “Recent” has been used in several places for papers that were published many years ago, such as the ones in lines 62 and 63. Methods. Again this section is too long and could be considerably shortened. Why was the literature search confined to papers published after 1990? Table 1 and Table 2 are cited in reverse order in the text – Table 2 in line 119 and Table 1 in line 136. Results. It is not necessary to have both Tables 3 (covering 7 pages) and Table 4 (covering 6 pages) as well as very full accounts of each study in the text. Reference to conference abstracts are not permitted in a review and should be omitted. It is notable that the included studies were published in 2000, 2005 (abstract), 2000, 2001, 2005, 2009 and 2019 (abstract). Therefore, apart from the last in this list, they are certainly not recent, a fact that deserves comment as attitudes towards personal sun exposure have changed since around 1990 in several countries, such as Australia and New Zealand. It would be of interest to add latitude and mean UV Index in the summer and winter months in each study outlined in Table 3. Discussion. Again this section is too lengthy with little criticisms of the included publications and what type of study would be required to address the aim of the review. Discussion relating to some recent references is required such as Tabbakh T et al (PLoS Med 2019, 16, e1002932) and Wright B et al (Health Promot J Austr 2019, 30, 267-71).

Author Response

Response to reviewer 2

We would like to thank the reviewer sincerely for providing us with valuable comments and suggestions. We feel that these have enabled us to further optimize the paper. Our response to the indicated comments are below. We genuinely hope that the reviewer finds the revised manuscript suitable for publication in the International Journal of Environmental Research and Public Health.

Comment

‘As only seven studies were judged relevant for inclusion by the authors, the review is too long at 35 printed pages and is repetitive in places. It could be considerably shortened, thereby making more impact.’

Response: Thank you for your comment. We agree with the fact that the manuscript is elaborate. This is largely due to the fact that we have strictly adhered to the PRISMA guidelines for reporting on and conducting systematic reviews. By adhering to this guidelines, we have aimed to increase methodological transparency, which is highly important for reproducibility of the study. Moreover, in different research fields, following the PRISMA statement demonstrated increased methodological quality and quality of reporting (e.g. Panic et al., 2013; Sharma, et al., 2018). Even though this has caused detailed, transparent and therefore sometimes elaborate descriptions of methodology and results sections, we strongly believe in its importance and the positive impact this may have on the quality of this manuscript within our specific research field. Nevertheless, we have reduced parts of the manuscript whenever suitable.

Comment

‘Introduction. This is not sufficiently focussed. Explaining details of dual process models in the fourth paragraph is not appropriate, for example. References in the first paragraph need to be updated. Mistakes in English are frequent such as in line 29 where “one” should be “two”, in line 47 where “recreating” should be “recreation”, “proportional time” in line 48 does not mean anything, nor does “proportional amount” in line 63. What does “micro” or “macro” level mean in line 86? In line 90, the guidelines are those advised by WHO; the UV Index threshold of 3 has been omitted. “Recent” has been used in several places for papers that were published many years ago, such as the ones in lines 62 and 63.’

Response:

We have updated some references in the first paragraph and have restricted the paragraph in which we exemplify different dual process models within social psychology. Furthermore, we adjusted the English grammar/spelling where needed. Additionally, we have emphasized that sun protection guidelines described in the report of the Surgeon General are based upon recommendations from the WHO. With regards to your question about the micro or macro level in line 86; we explain the meaning of these concepts in the second paragraph, in which we outline dual process models and various types and levels of influence on individuals. The usage of different levels of influence is explained in ecological models of health behavior and we believe interventions targeting the physical environment can be best-distinguished making use of these levels of influence.

Comment

‘Methods. Again this section is too long and could be considerably shortened. Why was the literature search confined to papers published after 1990? Table 1 and Table 2 are cited in reverse order in the text – Table 2 in line 119 and Table 1 in line 136.’

Response: We understand the length of the method section is quite excessive. However, we decided upon an extensive method section since defining the systematic search process is important. Moreover, we adhered to the PRISMA guidelines and checklist for reporting Systematic Reviews, which states (in the method section specifically) that all study characteristics should be reported on in detail. Furthermore, we decided on 1990 as cut-off point for including studies since the early nineties present a starting point for sun safety interventions, which for example is demonstrated in the launch of the Sun Smart program in Australia around this time. Moreover, since we were particularly interested in effect evaluations of environmental components of such interventions, we do not believe earlier studies conducted would provide us with relevant studies adhering to our research questions.

Comment

‘Results. It is not necessary to have both Tables 3 (covering 7 pages) and Table 4 (covering 6 pages) as well as very full accounts of each study in the text.’

Response: We understand your suggestions concerning elaborative descriptions of the Results section. We have, as we mentioned before, chosen to report findings extensively as we closely followed the PRISMA guidelines since we strongly believe in its importance within our research field. Furthermore, we can add the two tables regarding study characteristics and study outcomes as supplementary files, however we believe all aspects depicted in these Tables are relevant for providing an overview for readers.

Comment

‘Results. Reference to conference abstracts are not permitted in a review and should be omitted.’

Response: We understand your suggestion. However, we believe including conference abstracts in systematic reviews is relevant. Moreover, using grey literature such as these types of research is being recommended for systematic reviews (e.g. Paez, 2017). Based on your comment, we have added a short reflection on this topic in the strengths and limitations paragraph in the discussion, though, in order to justify this decision.

Comment

‘Results. It is notable that the included studies were published in 2000, 2005 (abstract), 2000, 2001, 2005, 2009 and 2019 (abstract). Therefore, apart from the last in this list, they are certainly not recent, a fact that deserves comment as attitudes towards personal sun exposure have changed since around 1990 in several countries, such as Australia and New Zealand. It would be of interest to add latitude and mean UV Index in the summer and winter months in each study outlined in Table 3.’

Response: Thank you for your comment. We agree that most studies included in our review are not very recent. We have added a short reflection on this matter in the strengths and limitations paragraph in the discussion. Furthermore, we acknowledge the fact that different countries (and thus different latitudes and mean UV-indexes) ask for different skin cancer prevention strategies. Further, changes in attitudes or societal norms can certainly be of influence in sun exposure behaviors, especially in countries where skin cancer is a prolonged problem, such as in Australia. However, to correct for these types of influences, in this review we have purposely examined the stand-alone effects of environmental interventions by comparing control and intervention groups. Attitudinal changes concerning UVR exposure, for example, should therefore not confound results regarding environmental adaptations.

We agree on the proposition that latitudinal differences between countries are interesting to take into account. Therefore, we have added a paragraph in the discussion section about latitudinal differences between countries and the need for sun protection strategies.

Comment

‘Discussion. Again this section is too lengthy with little criticisms of the included publications and what type of study would be required to address the aim of the review. Discussion relating to some recent references is required such as Tabbakh T et al (PLoS Med 2019, 16, e1002932) and Wright B et al (Health Promot J Austr)’

Response: We have added a short reflection on the publication year and total number of included studies in the strengths and limitations section of the discussion. Moreover, we have added a recommendation that future research is necessary to consolidate the findings of this review. With regards to your suggestion of including recent references; we have considered the papers you mentioned regarding effectiveness of sun smart programs in Australia. These papers describe multi-component programs in which environmental components are part of the intervention. Since our review specifically focused on the isolated effects of environmental interventions, we believe the discussion should focus on these characteristics exactly. 

Round 2

Reviewer 1 Report

Thanks for making changes to the manuscript.

Author Response

On behalf of all authors, thank you very much for your respons. 

Karlijn Thoonen

Reviewer 2 Report

Although the authors have marginally improved their manuscript on revision, it is very disappointing that the length, rather than being substantially shortened as recommended, is now longer by 6 pages.  

Only a small portion of the Introduction has been omitted with the other sections being substantially unchanged or even lengthened. As the authors themselves recognise the “quite excessive length” of the Methods section, and “elaborative description” in the Results section, it is impossible to understand why this issue was not addressed. Suggestions were made about areas of the article which could be briefer but these were not accepted by the authors. Conference Abstracts are not peer-reviewed and frequently contain preliminary uncorroborated data, so it would be very unusual to include the information from such sources in a systematic review.   This is particularly true of the Abstract published in 2005, which, as there has been no follow-up regular paper in the succeeding fourteen years, calls into question the validity of the data in the original Abstract. The English remains at a poor standard. For example, the Abstract, which should be clear and unambiguous, is difficult to comprehend in its present form. The almost total omission of discussing recently published articles which contain parts relevant to the current study is hard to understand.

Author Response

Reviewer 2.

Commentary

Although the authors have marginally improved their manuscript on revision, it is very disappointing that the length, rather than being substantially shortened as recommended, is now longer by 6 pages. Only a small portion of the Introduction has been omitted with the other sections being substantially unchanged or even lengthened. As the authors themselves recognise the “quite excessive length” of the Methods section, and “elaborative description” in the Results section, it is impossible to understand why this issue was not addressed. Suggestions were made about areas of the article which could be briefer but these were not accepted by the authors. Conference Abstracts are not peer-reviewed and frequently contain preliminary uncorroborated data, so it would be very unusual to include the information from such sources in a systematic review. This is particularly true of the Abstract published in 2005, which, as there has been no follow-up regular paper in the succeeding fourteen years, calls into question the validity of the data in the original Abstract. The English remains at a poor standard. For example, the Abstract, which should be clear and unambiguous, is difficult to comprehend in its present form. The almost total omission of discussing recently published articles which contain parts relevant to the current study is hard to understand.

Response:

We thank the reviewer for his/her valuable comments. We did our best to fully address these. We acknowledge the length of the manuscript and have therefore done our best to further shorten the manuscript, resulting in a decrease of 7 pages in total. The one comment we did not fully adhere to was the exclusion of the conference abstracts. Since inclusion of conference abstracts specifically is permitted or even considered worthwhile in systematic reviews (Paez, 2017; Scherer & Saldanha, 2019) and we consider the inclusion of the abstracts in our systematic reviews as valuable and important, we strongly believe that these should not be omitted. Moreover, based on considerable contact with the authors during the study selection process, which we point out in the results section, we concluded the studies conducted are of good quality. Besides, the findings in the two conference abstracts were based on published baseline results or a protocol study which were both of strong methodological quality. By contacting the authors, we gained insight in relevant methodological aspects and details about follow-up results, which enabled us to estimate the study quality properly. Additionally, with regards to the comment the reviewer made about the conference abstract published in 2005; the abstract we included dates from 2010. We have indicated this more clearly now. Lastly, we integrated the study of Tabbakh (et al, 2019) in our discussion as the reviewer first suggested, since we acknowledge the importance of this particular study in the scope of effectiveness of multi-component sun safety interventions.

References

Paez, A. (2017). Gray literature: An important resource in systematic reviews. Journal of Evidence‐Based Medicine, 10(3), 233-240.

Scherer, R. W., & Saldanha, I. J. (2019). How should systematic reviewers handle conference abstracts? A view from the trenches. Systematic reviews8(1), 264.

Tabbakh, T., Volkov, A., Wakefield, M., & Dobbinson, S. (2019). Implementation of the SunSmart program and population sun protection behaviour in Melbourne, Australia: Results from cross-sectional summer surveys from 1987 to 2017. PLoS medicine, 16(10).